# Empowering Women: Moving from Awareness to Action at the Immunology of Fungal Infections Gordon Research Conference

**DOI:** 10.3390/pathogens8030103

**Published:** 2019-07-17

**Authors:** Elizabeth R. Ballou, Sarah L. Gaffen, Neil A. R. Gow, Amy G. Hise

**Affiliations:** 1School of Biosciences, University of Birmingham, Edgbaston, Birmingham B15 2TT, UK; 2Institute of Microbiology and Infection, University of Birmingham, Edgbaston, Birmingham B15 2TT, UK; 3University of Pittsburgh, Division of Rheumatology and Clinical Immunology, Pittsburgh, PA 15261, USA; 4School of Biosciences, University of Exeter, Geoffrey Pope Building, Exeter EX4 4QD, UK; 5Department of Pathology, School of Medicine, Case Western Reserve University, Cleveland, OH 44106, USA; 6Louis Stokes Cleveland VA Medical Center, Cleveland, OH 44106, USA

**Keywords:** gender disparity, unconscious bias, women in science, women in medicine

## Abstract

Despite the high prevalence of women in graduate degree programs and equal or more women earning PhDs, MDs, and MD/PhDs, and despite efforts at individual and institutional levels to promote women in STEM fields, there remains a disparity in pay and academic advancement of women. Likewise, there is a paucity of women in top scientific and academic leadership positions. The causes of this gender disparity are complex and multi-factorial and to date no “magic bullet” approach has been successful in changing the landscape for women in academic and scientific fields. In this report we detail our experiences with a novel mechanism for promoting discussion and raising awareness of the challenges of gender disparity in the sciences. The Gordon Research Conferences (GRC) launched the Power Hour at its meetings in 2016: a dedicated, scheduled session held during the scientific meeting to facilitate discussion of challenges specific to women in science. Here we share our experience with hosting the second Power Hour at the 2019 GRC Immunology of Fungal Infections (IFI) meeting held in Galveston, TX. We will discuss the overall structure, key discussion points, and feedback from participants with the aim of supporting future efforts to empower women and underrepresented minority groups in science.

## 1. Introduction

170 years ago, Elizabeth Blackwell became the first woman to graduate from medical school when she finished at the top of her class at Geneva Medical School in Geneva, N.Y. Twenty-eight years later in 1877 Helen Magill White was the first woman to earn a PhD in the United States (studying Greek at Boston University). Today women make up nearly half of US medical school classes and account for the majority of graduate degrees and certificates awarded [1,2]. Even so, the representation of women in academic and industry scientific leadership has lagged behind. For example, in 2018 in the US there were only 27 women deans of medical schools, making up 18% of decanal positions, yet women comprise 41% of all full time medical school faculty [3]. Similarly, across 110 countries for which data is available, nearly half (44%) of Science, Technology, Engineering, and Math (STEM) graduates are women, but representation declines steadily throughout career progression, with fewer than 20% of full professors being women across disciplines and across nations [4,5,6,7]. Depending on the field of science, university and institutional leadership by women is typically much lower than 20%, although the full extent of this deficit remains an area of active study [8].

Gender bias in grant review has also been documented; this bias disappears when proposals are judged solely on their scientific merit, likely reflecting unconscious bias in addition to gender discrimination [9,10]. There are also discrepancies in hiring, pay, and academic rank between women and men, particularly at the higher ranks [1,11]. Even among those with advance degrees in the US, women are paid 74% less than men [12]. Women also receive tenure at a disproportionately lower rate and are less likely to find employment in their field of study [1,13].

While a common perception has been that this gender discrepancy will shrink as more women enter STEM fields, a gap persists in the percent of women researchers who successfully make the transition to independence. A 2018 survey of principle investigators (PIs) in the UK who had launched their labs in the preceding 7 years found that similarly qualified women PIs were still paid less than men at a comparable career stage, translating to £3–5k, or 10%, difference in wages at the start of their career [14]. This was reflected in the lower starting grade (lecturer vs. senior lecturer) at which the majority of women were appointed, further limiting the rate of their subsequent career progression relative to men in their cohort. A consequence of even a small percentage difference in starting pay magnifies over the course of time, as pay raises are often given as a percentage of current salary.

In addition to the pay and promotion gaps, women worldwide face challenges in accessing resources within their institutions [15]. Prof Nancy Hopkins documented widespread marginalization of women researchers at MIT in the 1990s, including allocation of lab space and funds for small equipment as well as promotion to leadership roles [16]. Her findings led to the establishment of a Committee on Women Faculty that was able to improve outcomes for women at MIT [16]. However, as Acton et al. demonstrate, this problem persists: a 2018 survey of new PIs in the UK found that new women PIs are systematically less well-resourced by their hiring departments than their male counter-parts, further contributing to the achievement gap. This was reflected in reduced grant capture and smaller group sizes in the first 5 years when the PIs were women [14]. This can have long term effects, as access to resources will impact on a researcher’s ability to address tenure and promotion criteria include grant capture, the generation and publication of research, and markers of external recognition of esteem (invited seminars, awards, participation on grant review panels, etc.) [17].

Within their broader research communities, women in STEM fields continue to face gender discrepancies, for example in representation as invited speakers at meetings. A 2014 survey of Academic Grand Rounds found that, despite women comprising 46.7% of medical students, women speakers at Grand Rounds comprised only 26.2% (median) and that this was lower for invited external women speakers (22.4%, median) [18]. This trend could be observed for all specialties except obstetrics/gynaecology and surgery, and a similar gap has been identified at academic research conferences across disciplines [19,20,21,22,23]. In addition to combating stereotypes, speaking invitations enable research dissemination, allow access to travel bursaries, and can lead to further invitations for community leadership. Such roles are markers of external recognition, and as such are key metrics for grant proposals, tenure, and promotion.

A common perception is that under-representation of women as invited speakers is due to a lack of qualified women, where impact factor is used as a proxy for quality [24]. Klein et al., (2017) set out to specifically test this idea in the field of neuro-immunology, where a lack of gender balance (<50% women) at national and international meetings in 2016 was observed in 66% of preliminary programs [23]. Using publication impact factor in the preceding two years as a proxy for quality, these authors demonstrated that, contrary to perception, invited women speakers (15.2% of invited speakers) were on average more qualified than invited men speakers, and that a proportion of invited men speakers (21%) were less qualified than un-invited women speakers identified by an expert committee comprised of both women and men. When presented with a list of qualified women researchers who had not been selected, organizers identified additional speakers who were subsequently invited.

It should be noted that the use of impact factor has been shown to be a poor proxy for quality that can embed gender bias into an analysis [25]. For example, Klein et al. specifically focused on first and last author publication impact factors [23]. However, even in cases of joint authorship, where both authors are stated to have contributed equally, women are less likely to be listed as first [26]. Therefore, this approach may have underestimated the quality of women speakers.

Overall, despite significant efforts on the part of research and academic institutions, scientific societies, group leaders, and individual scientists, there still exists significant gender inequity in science and medicine. This is a complex and multifactorial problem that reflects institutional and individual biases, overt and subtle gender discrimination, differential societal and family roles and commitments, and different career trajectories, among other issues.

## 2. Steps the GRC has Taken to Raise the Profile of Gender Issues

Halpern et al.’s landmark meta-analysis demonstrated in 2007 that differences in STEM achievement are linked to culture and environment, and Acton et al. found that the support of mentors correlates with a more optimistic view of the future, particularly among women [14,27].

Recognizing the need to take concrete steps to raise the profile of women in science, the Gordon Research Conference (GRC) launched the Power Hour at its meetings in 2016 [28]. The aim of this program is to create a forum for attendees to discuss challenges specific to women in science and to support their professional development through discussion and mentoring.

In 2017, the GRC held 108 Power Hours across their conferences, including at the Immunology of Fungal Infections (IFI) meeting. At the 2017 IFI Power Hour, there were approximately 50 attendees (~25% of the total attendees) with 75% women and 80% graduate students and post docs (personal communication, GRC Administrator). In 2019, the 2nd IFI Power Hour, which we led and organized, drew 46 participants, majority female (83%), spanning all career stages from graduate student to full professor. Here we share our experience with this event, including overall structure, key discussion points, and feedback from participants on impact and potential for improvement, with the aim of supporting future efforts to empower women and underrepresented minority groups in science.

## 3. Gender Representation at the IFI over Time

The IFI is a relatively new GRC meeting, first launched in 2011 to bring together disparate groups of researchers in fungal immunology and fungal pathogenesis. The goal of the meeting is to facilitate cross-disciplinary collaboration to drive basic research and the development of new antifungal therapies. Meeting attendees are encouraged to build networks that extend beyond the traditional boundaries of their disciplines. The structure of the meeting supports this by providing designated discussion time following each talk, substantial unstructured time during the day to enable collaboration, and a robust poster session preceded by flash talks from selected poster presenters. Since 2011, there have been a total of five meetings, each comprised of nine sessions with two to eight speakers per session, plus poster flash talks [29].

The 2019 meeting was organized by Chairs Sarah Gaffen and Neil Gow and Vice Chairs Ilse Jacobsen and Jatin Vyas, representing an even gender representation. The meeting attracted 195 participants from 10 countries and a range of career stages (Figure 1). The majority came from academic institutions, but industrial and government scientists also attended. The gender breakdown of the meeting was 48% female with 42% of the overall speakers and 44% of the discussion leaders being female, reflecting the composition of attendees and enabling overall visibility of women at the meeting (Figure 2 and Figure 3). This was achieved by Chairs and Vice-Chairs working to select speakers for the various sessions based on expertise and relevance while also committing to maintaining balance in gender, career stage, and geography, among other factors. Adherence to this approach was reinforced by policies from funders supporting the meeting, specifically NIH/NIAID. A decision was also made not to repeat any speakers from the 2017 meeting to encourage a diversity of perspective. 

A large-scale analysis of gender breakdown at meetings held by the American Society for Microbiology from 2011 to 2013 revealed significant inequality in speaker selection, leading to calls to improve gender balance in selected and invited speakers and better representation of the overall make-up of the research community [19]. By presenting data on the lack of gender parity at previous meetings and by encouraging the inclusion of women in organizing roles in their meetings in 2014 and 2015, the ASM achieved gender parity between attendees and speakers within two years [17]. 

Whereas at the ASM meetings, the gender of conveners was perceived to correlate strongly with gender parity, for the GRC IFI there appears to be limited correlation between conference organizer and gender breakdown of speakers or discussion section leaders (Figure 2). In addition, there has been an overall increase in gender parity over the history of the meeting regardless of organizer gender breakdown, including a significant improvement in 2015 by the all-male Chairs of that meeting. Since 2015, the ratio of men and women speakers has been maintained through the subsequent meetings (Figure 3). The ASM likewise observed that since 2015 the need for women conveners to maintain gender parity among invited speakers appeared to disappear [17]. Based on our observations at the Power Hour and throughout the meeting, we believe the representation at this meeting is a reflection of the overall commitment of this community to increasing gender parity, including active consideration of gender balance in speaker selection and use of community input to identify a wide range of qualified speakers. For example, the Women Researchers in Filamentous Fungi and Oomycetes (WRIFFO) GoogleDoc, launched in April 2016, serves as a centralized source for identifying potential women speakers within our research community [30]. This mirrors the findings of Klein et al., indicating that if conference organizers make inclusivity a priority at community meetings, gender parity results [23]. 

Despite the progress our community has made on this front, individuals continue to face gender-based challenges in their professional lives. On the first day of the IFI meeting, all participants regardless of gender identity were invited to attend the Power Hour to discuss strategies for combatting gender bias.

## 4. Progress the Fungal Community has Made

Many participants at the IFI were also present at the 2018 Gordon Research Conference on Molecular and Cellular Fungal Biology (MCFB). Findings from the 2018 MCFB Power Hour were disseminated through publication and were used as a starting place for the IFI 2019 Power Hour [31]. Key discussion points raised at the 2018 MCFB Power Hour included unconscious bias, work-life balance, pay-gap balance, sexual harassment, and raising awareness. To build on their conclusions, the following topics were presented to the 2019 IFI Power Hour group for focused discussion:Identifying mentors and sponsors.Strategies for acting as an ally in cases of bias.Title IX and sexual harassment.Bias within academia (within the lab or within the department).Bias in the broader environment (administrators, reps, etc.).Bias encountered on the job market (interviews, negotiations, final offers).How do we recruit and retain under-represented groups to our field?

## 5. Our Experience with the Event

Following a short presentation of key statistics to the entire group (7 min), the topics were presented as possible themes to be discussed in small teams comprised of female and male junior and senior participants (5–7 people, 30 min). A recorder was selected for each team to encourage active listening and to facilitate reporting back to the whole group (15 min). Teams were instructed to focus on identifying strategies to address specific challenges, rather than sharing anecdotes about personal struggles. The organizers circulated through the groups to prompt discussion and refocus on problem solving where needed, and participants were encouraged to continue conversations after the Power Hour.

Team discussions were wide ranging and constructive. In many cases, participants shared their personal experiences with a view towards finding solutions or providing mentorship. A majority of the groups chose to focus on identifying mentors and sponsors, strategies for coping with bias, and bias encountered on the job market and within academia. A key limitation of the Power Hour identified by participants and organizers was the time provided for team discussion: at the end of the designated 30 min, teams were reluctant to end their discussions and report to the group. Dynamic and engaged conversations continued for a further 10 min before being cut short by the organizers to enable larger group discussion.

**Key Findings:** Several central themes emerged from the team and larger group discussions during the Power Hour at the IFI meeting. These will be discussed in more detail below.

*Mentorship:* Based on discussions with established IFI participants (define as those who have attended at least three meetings), we observed significant awareness of the problem of gender disparity in science at the senior levels and a desire to reduce barriers and improve access to opportunities for junior scientists. Participants emphasized the importance of fostering diversity across the gender spectrum within our research community. These discussions also extended to efforts to promote broader engagement with and retention of under-represented minorities to our field. In particular, this was demonstrated by a number of senior researchers who self-identified as potential mentors for junior scientists in search of support. As one long-time attendee put it, “I know I speak for many senior people in saying I am completely available to help/advise/mentor junior people on their career paths,” and this sentiment was echoed by others over the course of the meeting. Attendees to the Power Hour recognized that there are many challenges encountered by early career researchers in general, and women in particular, to identifying mentors. These may include a lack of self-confidence necessary to approach potential mentors, a lack of understanding about what the boundaries of mentor-mentee relationship should encompass, a lack of clarity about what makes a good mentor, misconceptions about the responsibilities of the mentor or mentee, and a lack of understanding about the distinction between a mentor and a sponsor. 

*Self-Promotion***:** Participants in the Power Hour discussion identified difficulty with self-promotion and negotiation as potentially challenging areas for women researchers. Many individuals (both men and women) shared their experiences with self-promotion and several concrete suggestions emerged. These included engaging peer support to encourage confidence and gain insight into norms and expectations, using peers to help role-play specific negotiations or discussions, developing and practicing an “elevator speech”—i.e., a 3- or 5-min summary of who you are and what your major areas of research and accomplishments are; and a general agreement that negotiation and self-promotion skills are important and likely to be utilized in multiple areas of one’s scientific career.

*Unconscious Bias:* Another topic that emerged from the discussion was the prevalence of unconscious bias (we all have it) and the need for mentors and mentees to understand their own level of unconscious bias. An example provided of a situation where unconscious bias may arise was in the writing of letters of recommendation. Studies have shown that letters of recommendation for women are often shorter, contain “grindstone adjectives” (i.e., hard-working) whereas those for men contain more standout adjectives (i.e., best, most, top) and achievement words (performance, career, leadership, knowledge) [32,33,34]. Discussants also highlighted the potential for unconscious bias to influence reviewer comments during grant review. The group concluded that there was a need for the entire scientific workforce to receive unconscious bias training and regular updated training (for example with online modules). 

*Hiring practices:* The subjects of unconscious bias and negotiations led to a conversation about hiring practices. Discussants who had participated on hiring committees highlighted procedures for candidate selection that may introduce bias into the candidate pool, including the use of recruiters, recruiting from a pre-selected pool, and the perception that selection should be blind to gender if the aim is to recruit the best applicants [35]. To combat the gender gap at the faculty level, hiring committees should ensure that recruiters and committee members are fully aware of department priorities, including the recruitment of minority candidates, and that members receive training in unconscious bias. 

*Work-Life Balance:* Finally, conversations about balancing professional and home life, particularly as it related to family planning and childrearing, were a focus of discussion. Parental responsibilities are a key driver of exit from the work place for both women and men [36]. In this regard, both women and men participants highlighted the challenges faced by working parents to attend meetings and balance child-rearing duties with work responsibilities. Several participants raised the point that access to facilities for nursing mothers, flexible working hours to enable parenting, and paternal as well as maternal leave were central to their ability to raise a family while remaining in the work place. Senior researchers also spoke frankly about their decisions to have or not to have children and how this affected their career trajectory, for example by impacting their ability to attend conferences. A consensus conclusion of this conversation was the importance of flexibility and creativity on the part of host institutions and organizations to enable women to successfully navigate the early childhood years, and that although some improvements had been made (designated space for nursing or pumping that was not a restroom; flexible working hours; limiting faculty meetings to core hours, inclusion of paternal leave; support for nursing mothers or co-parenting at conferences), that barriers remain: women often had to specifically request or advocate for these changes, rather than these being led from the top of the organization.

## 6. A Need for Strategies and Practical Solutions

Following the group discussion and as the meeting progressed, junior researchers in particular asked for practical tips on how to find allies in their own environment.

*Mentorship vs. Sponsorship:* A central theme throughout the team and larger group discussions was the role of mentorship in participant success and well-being, echoing the findings of Acton et al. that, among new PIs, those lacking mentors had a more negative outlook overall, and this affected women more strongly than men [14]. However, discussants expressed uncertainty about how to identify mentors and maintain mentorship networks. To help address this, the following points were made:Mentors can take many forms. In general, mentors offer advice and guidance, but each mentor may be a source of support in a limited area. Identifying mentors with expertise in different areas will allow mentees to have high value conversations with the relevant mentor when an issue arises.Individuals should seek out mentors both within and outside their local institutes and at different career stages. Perspectives on common challenges can evolve over time or be dependent on particular local conditions. Identifying a range of perspectives can help avoid bias and potential conflicts of interest.In seeking out mentors, individuals should look for shared outlook and other commonalities, rather than focusing on traits such as gender.Mentorships can be formal (also known as coaching) or informal. For formal mentorships, mentees should identify specific goals to be discussed in scheduled one-on-one meetings and should be prepared to reflect on their own progress.Successful mentor-mentee relationships are characterized by open-ended questions that allow the mentee to identify blind spots or alternate solutions to common challenges, rather than providing out-of-the-box solutions.Peer mentorship can be a valuable resource both in terms of support and in terms of building trusted networks within cohorts.Sponsors are a distinct group of senior scientists that can act as champions, advocating on a junior researcher’s behalf. Sponsors may be less directly involved in advising, but can be influential in advocating for access to opportunities. The expectations of a sponsor, who is invested in your professional success, may be distinct from those of a mentor, who is invested in your personal success.

*Building and Maintaining Networks:* All young scientists face challenges in their career progression that can be disruptive to mentorship and support networks. For example, 67% of new PIs in the UK had undertaken an international move at least once during their career, and 75% changed department or institution for their first independent post [14]. This can put pressure on support networks and make identifying and maintaining relationships with mentors challenging. The following actions were suggested to help overcome these challenges.

## 7. Suggested Actions for Individuals

Engage with unconscious bias training to assess how you may be influenced by your own biases (https://www.aamc.org/initiatives/diversity/322996/lablearningonunconsciousbias.html).Seek out opportunities for networking and development, or professional coaching. For example, the National Postdoc Association offers courses and advice (https://www.nationalpostdoc.org/), the European Network of Postdoctoral Associations provides links to local Postdoc Associations (https://www.uc.pt/en/iii/postdoc/ENPA), and Nature Jobs offers guidance on identifying mentors and developing mentorship skills (https://www.nature.com/naturejobs/science/career_toolkit/mentoring)Identify yourself as a peer mentor or voice willingness to mentor junior researchers in particular areas.Identify yourself as an ally and actively advocate for women and other under-represented minorities. Some examples can be found in [37].Encourage junior researchers by modelling mentorship in local groups such as those held by UCSF’s Women in Life Sciences group [38].Add names to the Women Researchers in Fungi and Oomycetes spreadsheet [30].Seek out opportunities for mentorship training such as that described by Hund et al. [39] and Chopra et al. [40].Establish or join a peer mentorship group at your local institute or online. There are several examples of these already, including NewPI_Slack and UK_NewPI twitter and slack channels that can be accessed online.Seek out opportunities for interacting with potential mentors at meetings or through local networks. Speak with peers or supervisors who may be able to help identify potential mentors.

## 8. Suggested Actions for Future GRCs

Allow extended time in the schedule for the Power Hour, to enable more in-depth discussion. In the program structure, the Power Hour precedes a poster session, so junior researchers who are presenting in this session are disadvantaged.Hold a dedicated session prior to the main meeting to address various areas in career development specifically focused on women and under-represented minorities. This could integrate early career participants of the Gordon Research Symposium as well as attendees of the GRC proper. Possible topics identified by 2019 discussants include:How/when to say yes or no to new opportunities/chores.How to more effectively handle the situation when you are the one on the receiving end of the bias.How to be aware of and decrease our own biases.Ask attendees to hold Office Hours during the meeting, in the breaks or at meals, when they would self-identify as being willing to act as mentors one-on-one or to small groups to discuss specific issues. These could be led by established PIs, but could also be a chance for students and post-docs to identify as peer mentors. This strategy targets three goals: (1) identifying mentors and potential sponsors outside trainees’ institution; (2) identifying allies; and (3) developing leadership skills and confidence when facing problems themselves.

## 9. Suggested Actions for Future Power Hours

Model difficult conversations surrounding a hypothetical but plausible case that could occur in a lab. Ask discussants to consider the problem from different points of view (PI, grad student, post doc, etc.).Consider how best to encourage conference attendees, particularly those who may be reluctant to engage, to join the Power Hour.Consider how to better use technology to share discussant perspectives. Word clouds and anonymous response submission systems can enable participation and perspective sharing from less vocal members.Consider how future Power Hours can take a wider view of challenges around gender and equality, particularly given the expanding understanding of the breadth of human gender identity. Power Hour conveners should consider specific challenges encountered by these groups including inability to access resources, bullying, and harassment.Consider how intersectionality with other characteristics (race, nationality, language, age, etc.) may impact attendee participation in the meeting and in career progression.

## 10. Suggested Actions for the Broader Microbiology Community

At future conferences, consider running sessions addressing some of the identified challenges.Engage with reports and surveys about the challenges facing junior researchers [14,41].Engage with efforts to collect information about gender and minority representation in STEM worldwide [8].Provide a forum for discussing the importance of diversity.Provide family friendly support (lactation rooms, child care subsidies, reduced fees for partners or support, travel grants for partners/caretakers).Consider gender balance at meetings, both at the organizational and speaker level. The Women Researchers in Fungi and Oomycetes database can serve as a resource [30].Continue to build on successes in promoting gender parity across invited and selected speakers and consider strategies for broadening participation in other ways (during question periods, at poster sessions, etc.). For example, research has shown that women tend to ask fewer questions at seminars, but a moderator who selects a woman to be first questioner can increase the number of women who participate subsequently [42].Expand strategies for raising the profiles of under-represented minority scientists beyond gender to other groups.Advocate at the institutional level for improved support and infrastructure for researchers taking leave related to caretaking and for those returning to work after a career break. Examples of initiatives in place include the Athena Swan Charter in the UK [43] and the National Institutes of Health (NIH) Gender Inequality Task Force Report in the US [44].

## 11. Conclusions

In conclusion, we found that hosting the Power Hour as part of the 2019 GRC IFI meeting was an engaging and educational experience. Participants valued the discussion and overall were happy with the format. Concrete recommendations and feedback from our session as well as other Power Hour sessions can be used to improve and expand the discussion and mentoring of women in science at future GRC as well as other scientific society meetings. These meetings provide a valuable and under-utilized opportunity for intentional and scheduled opportunities for participants to engage in discussion, problem solving, mentoring, and career development skill building as an approach to levelling the playing field for women in STEM. Of course, no single program can tackle this complex and pervasive issue. We believe that multiple approaches are needed, including the engagement of individuals, mentors and principle investigators, institutional and society leadership. Here we specifically focus on the challenges facing women in science that limit career progression and full participation in research. It should be noted that under-representation occurs across multiple axes. While the specific focus of the GRC Power Hour is on women, other groups may also face many of the challenges identified, or face other challenges that could be mitigated by the strategies identified, such as the assessment of unconscious bias, through culture shifts, and through the identification of allies, mentors, and sponsors. The Power Hour is a model program that has been successful in recruiting participation by meeting attendees from a variety of backgrounds and academic and leadership levels. We found the experience of leading the program to be quite valuable and recommend that GRC as well as other scientific societies and meeting organizers consider initiating or expanding these dedicated sessions devoted to issues around disparity. 

## Figures and Tables

**Figure 1 pathogens-08-00103-f001:**
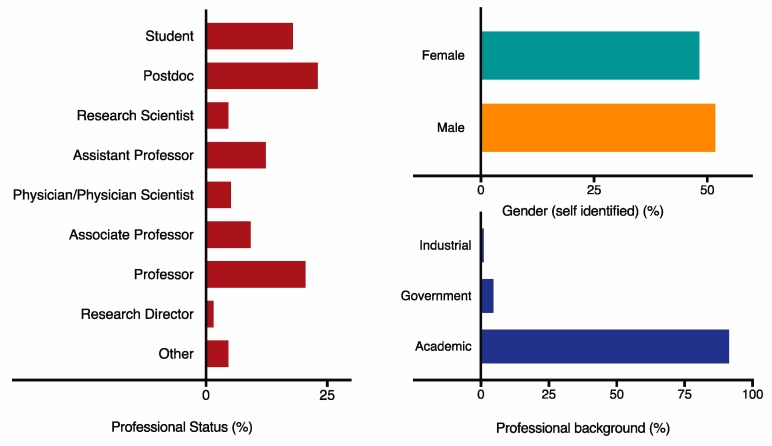
Demographics of the 2019 Gordon Research Conference Immunology of Fungal Infections (GRC IFI) Meeting. Self-reported demographic data were collected by anonymous voluntary survey after the GRC IFI Business Meeting, which was open to all conference attendees. Data are shown as percent, including gender breakdown, professional background, and professional status.

**Figure 2 pathogens-08-00103-f002:**
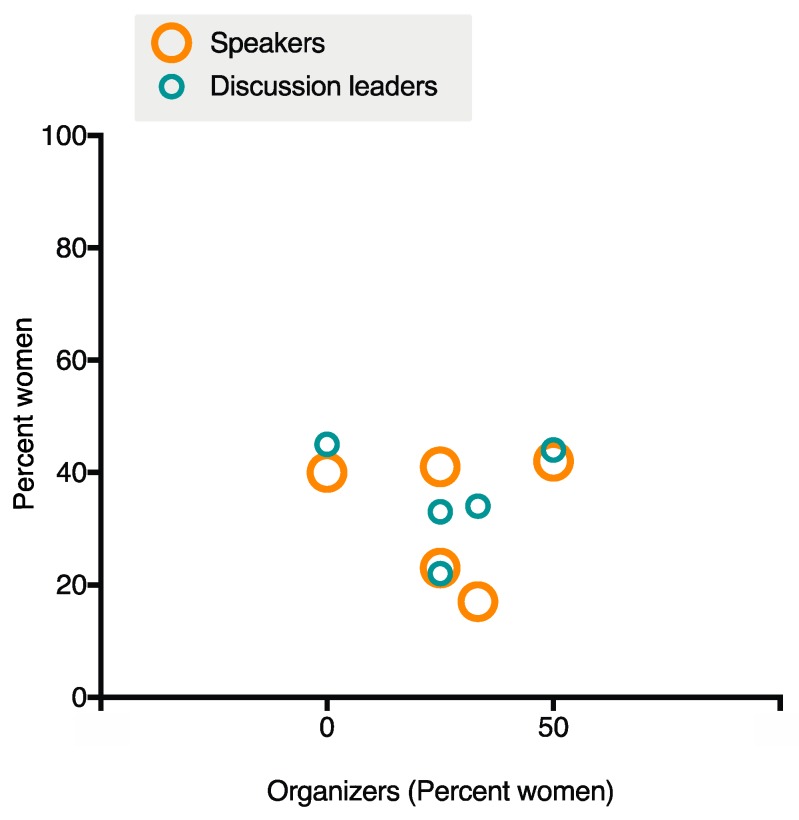
Relationship between Organizer Gender and Speaker or Discussion Leader Gender. Meetings had either one or two Chairs and two Vice Chairs. Gender breakdown is represented as percent women. There was no correlation between speaker or discussion leader gender breakdown and the presence or absence of women in Chair positions. Information about Meeting and Session Organizers and Invited and Selected Speakers was obtained from the GRC IFI meeting website for the last five meetings.

**Figure 3 pathogens-08-00103-f003:**
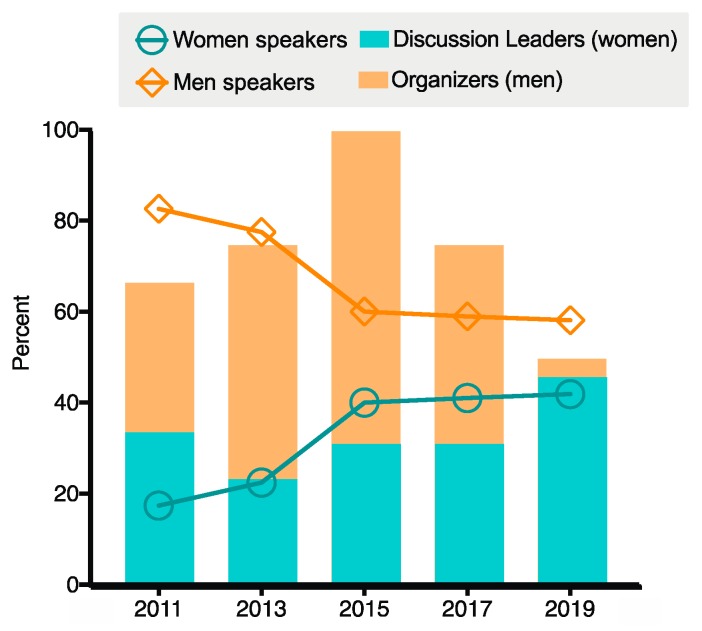
Total speaker gender breakdown over time. The gender breakdown of all invited and selected speakers for all five GRC IFI meetings is shown as a function of time. The gender breakdown of meeting organizers (presented as percent male, orange bars) is presented for each meeting. The gender breakdown of speakers (both male and female, orange and blue lines) is shown over time for each meeting. There has been a steady trend towards gender parity in speaker breakdown over the history of the meeting. The breakdown of discussion leaders selected by the organizers at each meeting is also shown (presented as percent female, light blue).

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
