# Peer review of "Empowering Women: Moving from Awareness to Action at the Immunology of Fungal Infections Gordon Research Conference"

_pathogens, 2019, doi:10.3390/pathogens8030103_

Round 1

Reviewer 1 Report

This manuscript by Ballou et al. adresses an important topic - the underrepresentation of women in top scientific and academic leadership positions - by describing the GRC Power Hours and sharing the results of the Power Hour held at the GRC Immunology of Fungal Infections in 2019.

It is well written, provides useful insights into mechanisms which have been show to promote equal gender representation, and  presents an accurate account of the IFI 2019 Power Hour (I was a particpiant myself). Furthermore, the results of the Power Hour present a useful reference of the key issues associated with the underrepresentation of women that is likely helpful for both individuals and institutions aiming to adress these issues. For individuals, the suggested actions list useful ideas and resources, such as websites and references. I furthermore think that the suggestions for future actions by the GRC and for the Power Hour are not only valid in this specific context but also for organizers of other meetings who want to incorporate a similar format.

I only have very few and minor points concerning formatting and clarity that I have highlighted in the attached document.

Reviewer 2 Report

This submitted report elegantly presents the discussions that formed part of the Power Hour at the Gordon Research Conference on “Immunology of Fungal Infections” in 2019. The authors not only provide feedback from the meeting but also include very informative context regarding the need to implement discussions about the advancement of women in STEM fields. The data presented here is very promising, specifically the data showing how the GRC IFI Chairs were able to close the gender disparity gap of speakers within 2 years (from 2013 to 2015). It is impressive that this gap has remained close for subsequent meetings. Furthermore, this report not only describes the progress made by the GRC IFI itself but also discusses progress made by the fungal community and is therefore valuable to a broader community of researchers. The strength of this report is the suggested actions going forward. These suggestions are directed at various levels ranging from individuals to the broader microbiological community, but also discuss specific action points for the GRC IFI itself. Overall this report was very well written, adds value to both the local fungal community and the broader microbiology community and raises very important points.

There were only a few minor comments/suggestions as follows:

The authors could consider broadening the stats discussed at the beginning of the introduction to beyond that of the US only. The authors could mention that there is very little data at the international level which shows the extent of gender inequality but that programs such as the “STEM and Gender Advancement” program from the Swedish International Development Agency (SIDA) will collect international data with the aim of developing better target policies (see link below).   

http://www.unesco.org/new/en/natural-sciences/priority-areas/gender-and-science/improving-measurement-of-gender-equality-in-stem/stem-and-gender-advancement-saga/

At the end of the introduction the authors describe some of the challenges faced by women being invited as speakers and conference chairs. However no mention is made of the challenges some women may face accepting invitations to speak, due to for example family responsibility. This is alluded to in the final action points for the broader microbiology community by the suggestion of family friendly support. In my experience, certain funders now require conference organizers to provide child care options for the duration of the conference.

In Figure 3 legend please clarify, “all five meetings”. Are the authors referring to the GRC IFI?
